# Enhancing Preoperative Outcome Prediction: A Comparative Retrospective Case–Control Study on Machine Learning versus the International Esodata Study Group Risk Model for Predicting 90-Day Mortality in Oncologic Esophagectomy

**DOI:** 10.3390/cancers16173000

**Published:** 2024-08-29

**Authors:** Axel Winter, Robin P. van de Water, Bjarne Pfitzner, Marius Ibach, Christoph Riepe, Robert Ahlborn, Lara Faraj, Felix Krenzien, Eva M. Dobrindt, Jonas Raakow, Igor M. Sauer, Bert Arnrich, Katharina Beyer, Christian Denecke, Johann Pratschke, Max M. Maurer

**Affiliations:** 1Department of Surgery, Campus Charité Mitte and Campus Virchow-Klinikum, Charité—Universitätsmedizin Berlin, 13353 Berlin, Germany; mariusibach@gmail.com (M.I.); christoph.riepe@charite.de (C.R.); felix.krenzien@charite.de (F.K.); eva.dobrindt@charite.de (E.M.D.); jonas.raakow@charite.de (J.R.); igor.sauer@charite.de (I.M.S.); christian.denecke@charite.de (C.D.); johann.pratschke@charite.de (J.P.); max-magnus.maurer@charite.de (M.M.M.); 2Hasso Plattner Institute, University of Potsdam, 14476 Potsdam, Germany; robin.vandewater@hpi.de (R.P.v.d.W.); bjarne.pfitzner@hpi.de (B.P.); bert.arnrich@hpi.de (B.A.); 3Department of Information Technology, Charité—Universitätsmedizin Berlin, 13353 Berlin, Germany; robert.ahlborn@charite.de; 4Einstein Center for Neurosciences Berlin, Charité—Universitätsmedizin Berlin, 10117 Berlin, Germany; lara.faraj@charite.de; 5BIH Charité (Digital) Clinician Scientist Program, Berlin Institute of Health at Charité—Universitätsmedizin Berlin, BIH Biomedical Innovation Academy, Charitéplatz 1, 10117 Berlin, Germany; 6Department of General and Abdominal Surgery, Campus Benjamin Franklin, Charité—Universitätsmedizin Berlin, 13353 Berlin, Germany; katharina.beyer@charite.de

**Keywords:** upper gastrointestinal surgery, risk prediction, machine learning, artificial intelligence, esophagectomy

## Abstract

**Simple Summary:**

Preoperative risk prediction prior to oncologic esophagectomy is crucial for assisting surgeons in accurate patient selection and patients in their informed decision making. A new risk stratification tool, the IESG prediction model, was recently introduced, categorizing patients into different risk levelsMachine learning is a subfield of artificial intelligence and may allow for a more accurate identification of patients at risk. Therefore, we evaluated the IESG risk model and compared its performance with ML models. We found that the IESG risk model provided an overall adequate risk estimation. However, ML showed better results in the accurate risk stratification of patients, demonstrating its potential as a novel and powerful approach for future patient assessment.

**Abstract:**

Risk prediction prior to oncologic esophagectomy is crucial for assisting surgeons and patients in their joint informed decision making. Recently, a new risk prediction model for 90-day mortality after esophagectomy using the International Esodata Study Group (IESG) database was proposed, allowing for the preoperative assignment of patients into different risk categories. However, given the non-linear dependencies between patient- and tumor-related risk factors contributing to cumulative surgical risk, machine learning (ML) may evolve as a novel and more integrated approach for mortality prediction. We evaluated the IESG risk model and compared its performance to ML models. Multiple classifiers were trained and validated on 552 patients from two independent centers undergoing oncologic esophagectomies. The discrimination performance of each model was assessed utilizing the area under the receiver operating characteristics curve (AUROC), the area under the precision–recall curve (AUPRC), and the Matthews correlation coefficient (MCC). The 90-day mortality rate was 5.8%. We found that IESG categorization allowed for adequate group-based risk prediction. However, ML models provided better discrimination performance, reaching superior AUROCs (0.64 [0.63–0.65] vs. 0.44 [0.32–0.56]), AUPRCs (0.25 [0.24–0.27] vs. 0.11 [0.05–0.21]), and MCCs (0.27 ([0.25–0.28] vs. 0.15 [0.03–0.27]). Conclusively, ML shows promising potential to identify patients at risk prior to surgery, surpassing conventional statistics. Still, larger datasets are needed to achieve higher discrimination performances for large-scale clinical implementation in the future.

## 1. Introduction

Esophagectomy is the only potentially curative treatment for patients with advanced stages of esophageal cancer, a condition ranking as the sixth leading cause of cancer-related death [1,2,3]. Notwithstanding notable advancements in surgical techniques, the risk of major postoperative complications and perioperative mortality remains substantial, reaching rates as high as 38% and 9%, respectively [4,5,6,7,8]. Beyond short-term perioperative consequences, these complications significantly worsen the overall survival and quality of life of affected patients [9,10,11,12,13,14,15]. Accordingly, accurate preoperative risk stratification is critical to achieve optimal patient selection and support decisions grounded in informed consent [14,15,16,17]. Previous research has identified several risk factors for postoperative complications and mortality after esophagectomy, including age, comorbidities, and the type and stage of cancer [18,19,20,21,22]. These factors have been used to develop comprehensive risk scores and have subsequently been embedded in risk calculators to aid surgeons and patients in their joint decision-making process [23,24,25,26]. Recently, a new risk model predicting 90-day mortality after oncologic esophagectomy was introduced. Using logistic regression analysis on the multicenter International Esodata Study Group (IESG) database, the IESG risk model intends to provide an easily accessible risk stratification system, assigning patients into five distinct risk categories from very low risk to very high risk [4]. However, external validation of this newly introduced risk stratification model is currently lacking.

Moreover, patient-related and disease-associated factors that contribute to overall surgical risk may be interconnected in a non-linear, higher-order complexity. Machine learning (ML) algorithms, a subdomain of artificial intelligence (AI), are a powerful means of processing complex, non-linear system dynamics, thus allowing for the identification of high-dimensional patterns that may exceed human capabilities and conventional statistical approaches in medical research [27,28,29]. In support of this argument, ML prediction models have already shown promising results in forecasting long-term oncologic outcomes following upper gastrointestinal surgery [30,31]. In contrast, their potential to accurately identify patients at risk prior to oncologic esophagectomy remains to be evaluated—particularly in comparison to established preoperative risk models.

The objective of this study was to validate the IESG risk model concerning 90-day mortality and to assess its performance in comparison with new approaches from the field of ML. The results may assist patients and surgeons in their preoperative risk evaluation and provide a novel approach for optimized patient selection prior to oncologic esophageal resections.

## 2. Materials and Methods

### 2.1. Setting and Study Population

This retrospective case–control study enrolled patients after oncologic esophagectomy between January 2009 and December 2021 from two tertiary centers affiliated with Charité University Medicine Berlin. Center 1 served as ML training cohort and center 2 as independent ML validation cohort. Only patients who underwent Ivor Lewis (IL) esophagectomy were included, while other forms of resection were not considered. The primary study endpoint was defined as 90-day mortality, with 30-day mortality as the secondary outcome. This study was conducted in accordance with the Declaration of Helsinki, and the work is reported in line with the STROCSS criteria [32]. Data protection consultation was performed by the institutional data security department to safeguard patient privacy and confidentiality. The study further adheres to the guidelines for Transparent Reporting of a Multivariable Prediction Model for Individual Prognosis or Diagnosis (TRIPOD+AI https://www.equator-network.org/reporting-guidelines/tripod-statement/, accessed 8 August 2024) [33].

### 2.2. IESG Risk Model Validation

For validation of the IESG risk model, patients were stratified into five risk groups (very low to very high risk) using the parameter-weighted scoring system, as described by D’Journo et al. [4]. To assess model accuracy, estimated and observed mortality were considered, and a 95% Wilson confidence interval (CI) was calculated [34]. For discrimination performance, the area under the receiver operating characteristic curve (AUROC), the area under the precision–recall curve (AUPRC), and the Matthews correlation coefficient (MCC) were employed. The results were calculated for the full dataset as well as for the validation cohort exclusively to allow for an independent comparison between IESG model and ML performances.

### 2.3. ML Training and Validation

Preoperative data for ML analysis comprised 48 parameters encompassing categorical, bivariate, and numerical factors (Appendix A), including preoperative standard laboratory parameters. The tumor stage was determined using either endoscopic ultrasound or computer tomography imaging techniques according to the Union Internationale Contre le Cancer (UICC, 8th edition) TNM (“tumor”, “nodes”, “metastases”) classification of malignant tumors [35]. Comorbidities and the revised Charlson Comorbidity Index (CCI) were derived retrospectively, as described by Quan et al. [36].

The training and validation cohorts were subjected to statistical analyses using Student’s t-test and the chi-square test, including Yates correction, using SciPy 1.0 [37]. Data preprocessing for ML model development encompassed the deletion of features with >50% missing values, high collinearity (>0.85), and single unique values. Missing values in the remaining features were imputed, the numerical features normalized, and the categorical parameters were one-hot-encoded using the Scikit-learn framework [38].

ML development and evaluation were then performed using two consecutive approaches. In the first stage, the training cohort was divided into an internal training group and a test group in a ratio of 80:20 for the analysis. In the second stage, the entire training cohort was used for model development, while the independent validation cohort served to test the model for its robustness. Model selection and hyperparameter optimization were performed in both stages, using 5- and 8-fold stratified cross-validation on the corresponding training groups. Three different feature selection methods were assessed as part of the grid search: the selection of all features, the top 25 features based on ANOVA f-values, or the features whose coefficients exceeded the median within a support vector machine. The following classifiers were trained: decision tree (DT), logistic regression (LR), linear support vector machine (l-SVM), support vector machine (SVM), gradient boosting machine (GBM), random forest (RF), and a multilayer perceptron as a neural network (NN). A schematic of the model development is provided in Appendix A.

The optimal combination of hyperparameters was subsequently used to re-train the models on 100% of the patients in the respective training group, leveraging the maximum available data to construct the model for each approach. To reduce the variance introduced by randomly initialized model weights and the partitions of individual samples, the hyperparameter optimization process was subjected to an additional 100 repetitions. Finally, each optimized model was tested on the designated validation group; hence, these were the 20% subgroup within the internal training cohort and the independent external validation cohort, respectively. Model evaluation was performed through calculation of the AUROC, AUPRC, and MCC, which were computed and visualized using the Scikit-learn framework [38]. Finally, the performance results between the internal and external models were tested for significance using the Mann–Whitney–U test.

The complete code is publicly provided at https://github.com/HPI-CH/PROPEL (Accessed: 8 August 2024).

### 2.4. Feature Importance Analysis

To investigate the impact of each individual parameter on the constitution and performance of the trained models, feature importance calculations using Shapley additive explanations (ShAP) analysis were conducted for all classifiers averaging over 100 seeds. ShAP investigates alterations in model performance after the removal of single parameters individually, hence offering a detailed understanding of each feature’s quantified contribution to the model’s performance [39].

### 2.5. Comparison between IESG Model and ML Classifiers

To facilitate a comprehensive head-to-head comparison between the IESG and ML models, the discrimination performances of both methods were separately measured using the independent validation cohort with a subsequent comparison of the resulting AUROCs, AUPRCs, and MCCs.

## 3. Results

### 3.1. Patient Baseline Characteristics

Detailed descriptions of the patient characteristics of both the training and validation cohorts are given in Table 1. A total of 552 patients met the inclusion criteria and were selected for analysis. The overall completeness of the datasets was 95.2%. The training cohort comprised 409 patients, mostly male (n = 329, 80.4%), with a mean age of 63.8 ± 10.1 years. Adenocarcinoma (AC) was the most prevalent type of cancer, accounting for 65.3% (n = 267) of patients in the training cohort. A total of 85.6% (n = 350) of the patients had undergone neoadjuvant therapy, with 51.8% (n = 212) receiving chemotherapy and 33.3% (n = 136) undergoing radiochemotherapy. According to the revised CCI, the most common comorbidity was chronic pulmonary disease, affecting 24.0% (n = 98) of the patients, followed by diabetes without chronic complications (n = 59, 14.4%). The majority of the subjects (n = 255, 62.3%) had an Eastern Cooperative Oncology Group (ECOG) stage 0 performance status. T3N+ tumors accounted for the largest proportion within the training cohort in the preoperative tumor staging. The 90-day and 30-day mortality rates were 4.6% (n = 19) and 2.4% (n = 10), respectively. The leading complications associated with 90-day mortality were sepsis, with a predominance of respiratory tract infections (n = 6; 25%) and hemorrhagic shock (n = 5; 18.6%). Of the patients who died within 90 days after surgery, 26.3% (n = 7) had already been primarily discharged from the hospital.

The validation cohort included 143 patients, with the majority of patients being male (n = 110, 76.9%, *p* = 0.44), with a mean age of 65.8 ± 9.9 years. The prevalence of CCI comorbidities was in line with the training cohort, with chronic pulmonary disease (n = 27, 18.9%, *p* = 0.26) and diabetes without chronic complications (n = 25, 17.5%, *p* = 0.46) as the most common preexisting conditions. The revised CCI showed no significant difference between the cohorts (*p* = 0.46). In accordance with the training cohort, most patients had an ECOG 0 performance status (n = 99, 69.2%, *p* = 0.22) and T3N+-stage tumors in the preoperative tumor staging. AC was consistently the most common histological type of cancer (n = 93, 65.0%, *p* = 0.07), and the majority of patients underwent neoadjuvant chemotherapy (n = 61, 42.7%, *p* = 0.3). The most frequent complications causing 90-day mortality were sepsis (n = 6, 46.2%) and myocardial infarction (n = 2, 15.4%). A complete overview of all patient characteristics and parameters is given in Appendix A.

### 3.2. IESG Risk Model Evaluation

Stratification of all 552 patients into five different risk groups, as described by D’Journo et al., resulted in the following distribution: 2.4% (n = 13) of patients were categorized as very high-risk, 8.0% (n = 42) as high-risk, 19.7% (n = 109) as medium-risk, 22.6% (n = 125) as low-risk, and 47.6% (n = 263) as very low-risk patients. The expected and observed mortality rates for each group are shown in Figure 1. In the very high-risk group, the observed mortality was 23.1%, thus exceeding the reported IESG model prediction (18.2%) but meeting the 95% Wilson CI (0.08–0.50). The observed and predicted mortality within the high-risk category showed a slight overestimation (13.6% vs. 8.9%), however, again meeting the 95% CI (0.07–0.28). Among the patients classified in the medium-risk category, the observed mortality closely met the anticipated rates predicted by the IESG model (5.5% vs. 5.8%; CI: 0.03–0.11). The low-risk group showed one deceased patient, thus differing from the expected mortality rate (1.0% vs. 3.0%); however, the group was well within the corresponding CI (0.0–0.04). The majority of patients were assigned to the very low-risk group, in which the observed mortality rate of 6.1% surpassed the predicted 1.8%. Nevertheless, these outcomes again remained within the Wilson CI (0.04–0.11). A detailed categorization of the patients, according to D’Journo et al., is shown in Appendix A.

### 3.3. Machine Learning Model Evaluation

The primary ML models developed on 80% of the training cohort with consecutive testing on the withheld 20% showed low AUROCs ranging from 0.47 (0.44–0.51; SVM) to 0.50 (0.47–0.54; LR) for 30-day mortality but considerably higher results for 90-day mortality, ranging from 0.57 (0.54–0.59; DT) to 0.75 (0.73–0.78; LR). The corresponding model AUPRCs ranged from 0.03 (0.03–0.04; LR) to 0.06 (0.03–0.09; DT) for 30-day mortality and were again higher when calculated for 90-day mortality, ranging from 0.14 (0.11–0.18; DT) to 0.20 (0.17–0.23; SVM). In accordance, the MCC results showed higher prediction performances for 90-day mortality ranging between 0.15 (0.12–0.19; GB) and 0.37 (0.34–0.39; LR) compared to 30-day mortality modeling, with corresponding results between 0.01 (0.00–0.03; GB) and 0.11 (0.08–0.13). All results of the internal validation are given in Appendix A.

The models utilizing the complete training cohort for development with subsequent testing on the independent external validation cohort achieved AUROCs ranging between 0.52 (0.51–0.53; DT) and 0.64 (0.63–0.65; RF) for 90-day mortality (Appendix A). The corresponding outcomes for the AUPRCs and MCCs fell within the range of 0.10 (0.10–0.10; l-SVM) to 0.25 (0.24–0.27; NN; Appendix A) and 0.01 (0.00–0.01; l-SCV) to 0.27 (0.25–0.28; RF), respectively. Overall, the RF classifier demonstrated the highest AUROC and MCC results, while, notably, the NN classifier showcased the highest AUPRC. Moreover, all classifiers significantly decreased in terms of the AUROC compared to the preceding internal analysis. In contrast, the AUPRCs (Figure 2) demonstrated a notable performance increase for the DT (*p* < 0.01), the LR (*p* < 0.01), the SVM (*p* < 0.01), and, most notably, the NN (*p* < 0.01), as shown in Appendix A. In alignment with the preceding internal analysis, all metrics revealed higher discrimination performances for 90-day than for 30-day mortality prediction. A detailed summary of all model outcomes after external validation is presented in Table 2. Additional model performance metrics, including the F1 score, are provided in Appendix A.

Finally, the AUROC, AUPRC, and MCC were computed for the IESG risk model to allow for a direct comparison to the performance of the external ML evaluation. The IESG model performance showed an AUROC of 0.44 (0.32–0.56), an AUPRC of 0.11 (0.05–0.21), and a MCC of 0.15 (0.03–0.27), featuring relatively broad confidence intervals (Table 2). As a result, four out of seven ML classifiers (LR, NN, RF, and SVM) exhibited superior discrimination performances across all metrics compared to the IESG risk model.

### 3.4. Feature Importance

A heatmap displaying the normalized ShAP factor weights is presented in Figure 3, with additional details provided in Appendix A. The international normalized ratio (INR) emerged as the pivotal model parameter, consistently showcasing the highest weight in six out of seven classifiers (GBM, LR, NN, RF, l-SVM, and SVM). Subsequent to INR, the CCI demonstrated high model impact, ranking among the top three parameters in five classifiers (GBM, LR, NN, RF, and SVM) and within the first ten features in the two remaining models (DT and l-SVM). Furthermore, the red blood cell count (RBC) was consistently identified as being among the five most impactful parameters in four classifiers (l-SVM, LR, NN, and SVM). Considering patient characteristics, ASA status, weight, and age emerged as the most relevant factors. In terms of plasma-based laboratory parameters, C-reactive protein, gamma-glutamyl transferase, and bilirubin were identified as highly impactful.

## 4. Discussion

This study aimed to validate the IESG risk model as a new risk stratification tool and to compare its performance to ML algorithms for the preoperative prediction of 90-day mortality following oncologic esophagectomy. Adequate preoperative patient selection based on accurate risk stratification is critical for improving surgical outcomes after major surgical procedures and to aid patients in their informed consent decision making. The 90-day mortality provides a comprehensive endpoint, as it considers complications and adverse events beyond primary discharge, thus ensuring a more holistic assessment [40].

The IESG risk model assigns patients to one of five risk categories, providing a 90-day mortality risk estimation for each group. Applied to our independent cohort, the overall patient allocation aligned closely with the previously reported IESG distribution, as <3% of the patients were of very high risk, <10% were of high risk, and 20% were of medium risk. A notable shift was observed among the very low-risk category, which was twice as high as the IESG development cohort. Conversely, the group of low-risk patients was considerably smaller. Ultimately, the IESG stratification tended to underestimate the risk for patients within the very high-, high-, and very low-risk groups, which has to be considered when providing these patients with a mortality probability prior to surgery. Notably, however, all estimations met the underlying confidence intervals. The IESG risk group stratification can, therefore, provide an intelligible and intuitive grading system for patients as part of their preoperative decision making. In contrast, its discrimination performance was lower than previously reported, revealing an AUROC of 0.44, an AUPRC of 0.11, and an MCC of 0.15 when tested on our validation cohort.

Considering potential non-linear interdependencies among demographic patient characteristics, comorbidities, and disease-related factors in shaping cumulative surgical risk, multiple ML models were trained and validated. Importantly, we highlight that AUROC alone can be inconclusive when facing high class imbalances, which are often present in medical prediction tasks. AUPRC and MCC consider precision and recall as positive and negative predictive values, thus allowing for a more elaborate model evaluation [41]. Despite being trained on a dataset comprising less than 10% of the IESG cohort, the majority of ML classifiers outperformed the IESG model in discrimination performance. Notably, this still held true when comparing both approaches on the same independent validation dataset, emphasizing the potential of ML as a novel and powerful approach for surgical risk prediction in esophageal surgery.

The optimal model achieved an AUROC of 0.75 when trained on the internal cohort only, thus representing a monocentric analysis consistent with the study design of most ML analyses in the surgical field to date. Predicting major complications defined as Clavien–Dindo >IIIa, Jung et al. found AUROCs ranging between 0.6 and 0.7 in esophagectomies [42]. Likewise, Zhao et al. demonstrated AUROCs between 0.65 and 0.76 in the prediction of postoperative anastomotic leakage; however, they also included postoperative parameters in their models [43]. These findings, however, were not confirmed in independent cohorts, and therefore they are prone to the risk of overfitting in single-center data. In our study, subsequent external testing demonstrated a considerable AUROC decrease to 0.64, which remains critical for clinical implementation. However, we highlight that increasing the sample size by only 20% using the complete training cohort already optimized the AUPRC in various classifiers significantly, thereby exemplifying both the overall robustness of the approach and the potential for considerable improvements with larger cohorts in the future.

“Explainable AI” describes the efforts to improve the transparency and comprehensibility of AI models to foster trust among future users. In fact, non-comprehensibility has been shown to be a major barrier for surgeons in applying risk assessment tools thus far in clinical practice [44]. Providing professionals with underlying feature importance can, therefore, strengthen trust and acceptance. The INR and CCI consistently emerged as the most influential factors contributing to the final models. This finding is consistent with prior clinical studies examining risk factors for perioperative mortality in esophagectomy [2,45]. Particularly noteworthy is the significant impact of the preoperative laboratory variables across all classifiers. Given their accessibility as part of routine preoperative preparation, these variables should be more extensively leveraged in predictive modeling efforts.

Overall, our results suggest that high-dimensional models can improve surgical risk prediction beyond conventional analyses and may aid in identifying individual patients at high risk. Our results are in line with familiar studies in the field of colorectal and gastric surgery, where ML models trained on larger register datasets show promising preoperative prediction results, even surpassing AUROCs of 0.80 [46,47]. Performing large-scale analyses on national and international register data may, therefore, evolve as the next step in esophageal cancer research. Moreover, additional parameters may further increase accuracy in the future.

The primary constraint of this study lies in its sample size. This poses a major challenge for data-driven pattern recognition analyses and the identification of intricate interrelationships with high-dimensional data, particularly in the presence of strong class imbalances. Furthermore, the retrospective design may have introduced qualitative and quantitative data implications. However, automated data extraction using previously described means was applied to ensure homogeneity and to mitigate manual errors, ultimately achieving high dataset completeness. Moreover, in this study, IL esophagectomy was performed using open, laparoscopic, and robotic-assisted strategies, potentially introducing an approach-related effect on the outcome. While the total number of chemotherapy cycles was taken into account, potential dose reductions within the individual cycles were not. Similarly, variations in radiation dose were not considered in the radiotherapy cases, which could have influenced the outcomes. Finally, the training and validation cohorts exhibited some disparities concerning patient characteristics, most notably with a strong but not significant trend regarding 90-day mortality as the primary endpoint, thus potentially impeding comparability among the groups. Nevertheless, the ML tools evaluated for clinical implication must be sufficiently robust to produce reasonable results across diverse patient collectives. Adjusting or merging the cohorts would have severely compromised the quality of the analysis and particularly increased the risk of overfitting, which would decrease adequate model assessment.

## 5. Conclusions

The IESG risk model provides an easily accessible assessment approach to categorize patients into ascending risk groups and to provide a general risk estimation per group. However, complex models incorporating multiple dimensions can further enhance discrimination accuracy to identify patients at risk. Future studies applying non-linear models on large-scale register data are needed to evaluate the full potential of supervised and unsupervised ML analysis techniques in this field. Applying metrics that consider class imbalances as well as external validation are of high importance to accurately evaluate new models in the field of AI for upper-gastrointestinal surgery.

## Figures and Tables

**Figure 1 cancers-16-03000-f001:**
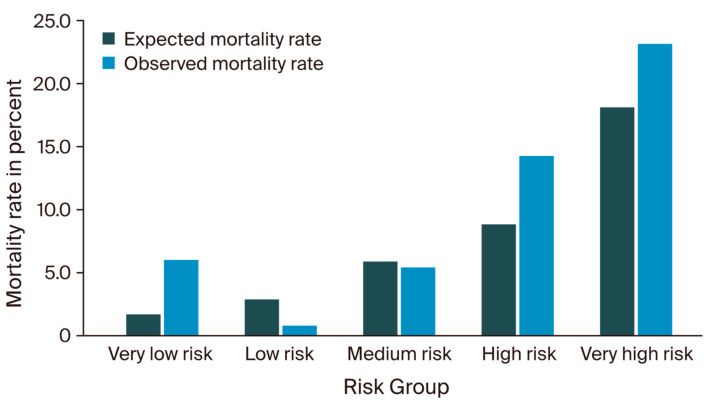
Expected versus observed mortality rates.

**Figure 2 cancers-16-03000-f002:**
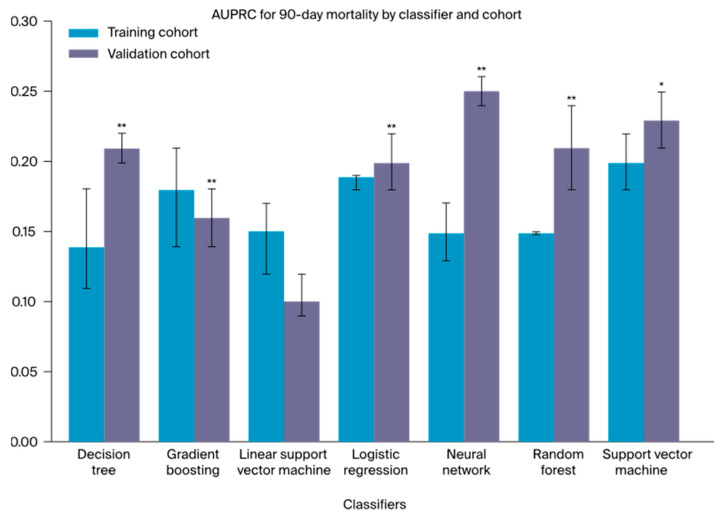
A one-sided Mann–Whitney U test was used to evaluate the difference in AUPRC between the training and validation cohorts of every classifier; * *p* < 0.05, significant; ** *p* < 0.01, highly significant.

**Figure 3 cancers-16-03000-f003:**
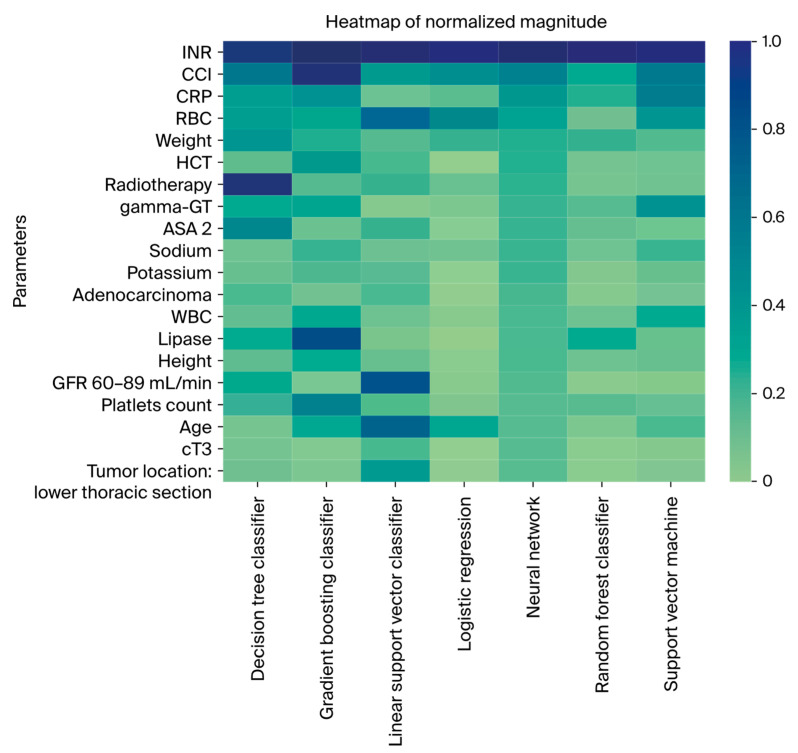
Heatmap displaying the ten most important features from ShAP value analysis for every classifier.

**Table 1 cancers-16-03000-t001:** Patient characteristics of the training and validation cohorts.

Characteristic	Total n = (552)	Training Cohort n = (409)	Validation Cohort n = (143)	*p* Value
Age				0.55
≤40 (%)	9 (1.6)	8 (2.0)	1 (0.7)	
41–50 (%)	42 (7.6)	33 (8.1)	9 (6.3)	
51–60 (%)	152 (27.5)	118 (28.9)	34 (23.8)	
61–70 (%)	178 (32.2)	130 (31.8)	48 (33.6)	
71–80 (%)	149 (27.0)	105 (25.7)	44 (30.8)	
>80 (%)	22 (4.0)	15 (3.7)	7 (4.9)	
Mean (SD)	64.3 (10.1)	63.8 (10.1)	65.8 (9.9)	
BMI				0.82
<18.5 (%)	22 (4.0)	17 (4.2)	5 (3.5)	
18.5–24.9 (%)	216 (39.1)	156 (38.1)	60 (42.0)	
25–29.9 (%)	240 (43.5)	182 (44.5)	58 (40.6)	
≥30 (%)	74 (13.4)	54 (13.2)	20 (14.0)	
Mean (SD)	25.8 (4.4)	25.8 (4.5)	25.7 (4.3)	
Sex				0.44
Male (%)	439 (79.5)	329 (80.4)	110 (76.9)	
Female (%)	113 (20.5)	80 (19.6)	33 (23.1)	
ECOG				0.22
0 (%)	354 (64.1)	255 (62.3)	99 (69.2)	
1 (%)	186 (33.7)	146 (35.7)	40 (28.0)	
2 (%)	12 (2.2)	8 (2.0)	4 (2.8)	
Histology				0.07
AC (%)	360 (65.2)	267 (65.3)	93 (65.0)	
SCC (%)	177 (32.1)	128 (31.3)	49 (34.4)	
Other (%)	14 (2.5)	14 (3.4)	0 (0.)	
Neoadjuvant treatment				0.3
Chemotherapy (%)	273 (49.5)	212 (51.8)	61 (42.7)	
Radiochemotherapy (%)	183 (33.2)	136 (33.3)	47 (32.9)	
Radiotherapy alone (%)	2 (0.4)	2 (0.5)	0 (0.0)	
Tpre				0.25
T0 (%)	1 (0.2)	1 (0.2)	0 (0.0)	
T1 (%)	42 (7.6)	30 (7.3)	12 (8.4)	
T2 (%)	67 (12.1)	54 (13.2)	13 (9.1)	
T3 (%)	300 (54.3)	246 (60.1)	54 (37.8)	
T4 (%)	23 (4.2)	19 (4.6)	4 (2.8)	
Tis (%)	1 (0.2)	1 (0.2)	0 (0.0)	
Tx (%)	8 (1.4)	4 (1.0)	4 (2.8)	
Npre				<0.001
N0 (%)	124 (22.5)	96 (23.5)	28 (19.6)	
N1 (%)	168 (30.4)	129 (31.5)	39 (27.3)	
N2 (%)	74 (13.4)	73 (17.8)	1 (0.7)	
N3 (%)	26 (4.7)	24 (5.9)	2 (1.4)	
Nx (%)	10 (1.8)	4 (1.0)	6 (4.2)	
Comorbidities				
CCI mean (SD)	7.4 (3.8)	7.3 (3.8)	7.6 (3.8)	0.46
Myocardial infarction (%)	19 (3.4)	13 (3.2)	6 (4.2)	0.76
Peripheral vascular disease (%)	24 (4.3)	13 (3.2)	11 (7.7)	0.04
Chronic pulmonary disease (%)	125 (22.6)	98 (24.0)	27 (18.9)	0.26
Peptic ulcer disease (%)	11 (2.0)	6 (1.5)	5 (3.5)	0.25
Liver disease mild (%)	32 (5.8)	19 (4.6)	13 (9.1)	0.08
Diabetes without chronic complications (%)	84 (15.2)	59 (14.4)	25 (17.5)	0.46
Hemiplegia or paraplegia (%)	12 (2.2)	7 (1.7)	5 (3.5)	0.35
Liver disease moderate/severe (%)	3 (0.5)	3 (0.7)	0 (0.0)	0.71
Renal disease (%)	2 (0.4)	2 (0.5)	0 (0.0)	0.98
Metastatic solid tumor (%)	154 (27.9)	115 (28.1)	39 (27.3)	0.93
30-day mortality (%)	15 (2.7)	10 (2.4)	5 (3.5)	0.71
90-day mortality (%)	32 (5.8)	19 (4.6)	13 (9.1)	0.08

AC = adenocarcinoma; SCC = squamous cell carcinoma; ECOG = Eastern Cooperative Oncology Group; CCI = Charlson Comorbidity Index; SD = standard deviation.

**Table 2 cancers-16-03000-t002:** AUROCs, AUPRCs, and MCCs for multiple ML classifiers after external validation for 90- and 30-day mortality, as well as AUROC, AUPRC, and MCC discrimination performance calculated for the IESG risk model (90-day mortality only).

	Classifier	AUROC Mean, 95% CI (Low–High)	AUPRC Mean, 95% CI (Low–High)	MCC Mean, 95% CI (Low–High)
90-day mortality	Decision tree	0.52 (0.51–0.53)	0.21 (0.17–0.24)	0.07 (0.05–0.09)
	Gradient boosting	0.64 (0.63–0.65)	0.16 (0.15–0.16)	0.12 (0.10–0.14)
	Linear support vector machine	0.51 (0.50–0.52)	0.10 (0.10–0.10)	0.01 (0.00–0.01)
	Logistic regression	0.64 (0.63–0.64)	0.20 (0.19–0.21)	0.26 (0.25–0.28)
	Neural network	0.61 (0.60–0.63)	0.25 (0.24–0.27)	0.24 (0.21–0.27)
	Random forest	0.64 (0.63–0.65)	0.21 (0.20–0.22)	0.27 (0.25–0.28)
	Support vector machine	0.62 (0.61–0,63)	0.23 (0.21–0.25)	0.21 (0.18–0.24)
	IESG score stratification	0.44 (0.32–0.56)	0.11 (0.05–0.21)	0.15 (0.03–0.27)
30-day mortality	Decision tree	0.49 (0.47–0.50)	0.17 (0.14–0.22)	0.03 (0.02–0.05)
	Gradient boosting	0.58 (0.56–0.60)	0.06 (0.06–0.07)	0.03 (0.01–0.05)
	Linear support vector machine	0.50 (0.49–0.51)	0.04 (0.04–0.05)	0.00 (0.00–0.00)
	Logistic regression	0.45 (0.44–0.45)	0.03 (0.03–0.04)	0.04 (0.03–0.05)
	Neural network	0.44 (0.41–0.46)	0.07 (0.06–0.09)	0.06 (0.03–0.08)
	Random forest	0.47 (0.45–0.49)	0.05 (0.03–0.06)	0.07 (0.05–0.08)
	Support vector machine	0.52 (0.51–0.54)	0.08 (0.07–0.09)	0.03 (0.01–0.04)

## Data Availability

The complete code is publicly provided at https://github.com/HPI-CH/PROPEL (accessed: 8 August 2024). The data can be requested from the corresponding author. After consultation with the relevant authorities, the data may be individually reviewed.

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
