# Peer review of "Enhancing Preoperative Outcome Prediction: A Comparative Retrospective Case–Control Study on Machine Learning versus the International Esodata Study Group Risk Model for Predicting 90-Day Mortality in Oncologic Esophagectomy"

_cancers, 2024, doi:10.3390/cancers16173000_

Round 1

Reviewer 1 Report

Comments and Suggestions for Authors

1. Significance and Originality of the Study

This study compares the performance of the conventional ISEG risk model with machine learning (ML) models for predicting 90-day mortality in esophageal cancer surgery. Risk prediction is crucial for assisting surgeons and patients in preoperative decision-making. The study is significant as it suggests the potential of ML as a novel approach in this context.

2. Research Methodology

  • Dataset: The study uses data from 552 patients collected from two independent centers, which is commendable. However, using a more diverse dataset could enhance generalizability.
  • Model Evaluation: Evaluating model performance using AUROC, AUPRC, and MCC is appropriate. However, there is a lack of detailed description of the model's hyperparameters and the specific training processes, raising concerns about reproducibility.

3. Results

  • ISEG Risk Model: The study confirms that the ISEG model is adequate for group-based risk prediction but underperforms compared to ML models.
  • ML Models: ML models achieved superior AUROC, AUPRC, and MCC compared to the ISEG model, indicating their promise. However, providing detailed reasoning for selecting the optimal ML model and explaining the specific performance differences between models would enhance understanding.

4. Conclusion and Future Prospects

  • Conclusion: The study suggests that ML models have the potential to surpass traditional statistical models, marking a step towards future large-scale clinical implementation.
  • Future Challenges: Although the need for larger datasets is mentioned, there is a lack of specific discussion on how to collect such data or improve prediction by incorporating additional variables.

Recommendations for Improvement

  1. Data Diversity: Including more diverse patient data can improve the model's generalizability.
  2. Model Details: Providing detailed descriptions of the hyperparameters and training processes for each ML model can enhance the study's reproducibility.
  3. Detailed Results Analysis: Conducting a more thorough analysis of the specific differences in performance metrics can clarify which models excel under what conditions.
  4. Additional Evaluation Metrics: Including other evaluation metrics (e.g., F1 score, recall) can provide a more comprehensive assessment of the models' performance.

By addressing these points, the study's contribution and reliability can be further enhanced.

Comments on the Quality of English Language

1. Significance and Originality of the Study

This study compares the performance of the conventional ISEG risk model with machine learning (ML) models for predicting 90-day mortality in esophageal cancer surgery. Risk prediction is crucial for assisting surgeons and patients in preoperative decision-making. The study is significant as it suggests the potential of ML as a novel approach in this context.

2. Research Methodology

  • Dataset: The study uses data from 552 patients collected from two independent centers, which is commendable. However, using a more diverse dataset could enhance generalizability.
  • Model Evaluation: Evaluating model performance using AUROC, AUPRC, and MCC is appropriate. However, there is a lack of detailed description of the model's hyperparameters and the specific training processes, raising concerns about reproducibility.

3. Results

  • ISEG Risk Model: The study confirms that the ISEG model is adequate for group-based risk prediction but underperforms compared to ML models.
  • ML Models: ML models achieved superior AUROC, AUPRC, and MCC compared to the ISEG model, indicating their promise. However, providing detailed reasoning for selecting the optimal ML model and explaining the specific performance differences between models would enhance understanding.

4. Conclusion and Future Prospects

  • Conclusion: The study suggests that ML models have the potential to surpass traditional statistical models, marking a step towards future large-scale clinical implementation.
  • Future Challenges: Although the need for larger datasets is mentioned, there is a lack of specific discussion on how to collect such data or improve prediction by incorporating additional variables.

Recommendations for Improvement

  1. Data Diversity: Including more diverse patient data can improve the model's generalizability.
  2. Model Details: Providing detailed descriptions of the hyperparameters and training processes for each ML model can enhance the study's reproducibility.
  3. Detailed Results Analysis: Conducting a more thorough analysis of the specific differences in performance metrics can clarify which models excel under what conditions.
  4. Additional Evaluation Metrics: Including other evaluation metrics (e.g., F1 score, recall) can provide a more comprehensive assessment of the models' performance.

By addressing these points, the study's contribution and reliability can be further enhanced.

Author Response

Dear Sir or Madame,

We would like to express our utmost appreciation for your valuable comments and wish to sincerely respond to your remarks accordingly.

  1. “Dataset: The study uses data from 552 patients collected from two independent centers, which is commendable. However, using a more diverse dataset could enhance generalizability.”

Thank you for your very valuable recommendation. We fully concur that using additional datasets could enhance the generalizability of our study. Although our current dataset from 552 patients across two independent centers already provides a solid foundation, we agree that incorporating data from a wider range of sources could further strengthen our findings. Inspired by your recommendation, we explored the feasibility of incorporating additional samples from the expansive open source MIMIC IV database, which encompasses 430,000 admissions. However, after rigorous data extraction, cleaning, and preprocessing of diagnoses, procedures, and parameters, the final cohort of additional esophagectomy cases comprised fewer than 130 eligible patients. Furthermore, the overall completeness of these identified samples was notably lacking, necessitating extensive imputation and ultimately compromising the quality of our analysis.

After thorough evaluation of these limitations, we are convinced  that adhering to our original cohorts provides a more appropriate, comprehensible, and transparent framework for conducting a head-to-head comparison between machine learning and the conventional ISEG score on our high-quality external cohort. However, based on your recommendation, we underscored the aspect of a more diverse dataset within the limitation section.

  1. “Model Details: Providing detailed descriptions of the hyperparameters and training processes for each ML model can enhance the study's reproducibility.”

We sincerely appreciate your feedback regarding reproducibility. In response to your recommendation, we have included the list of hyperparameters in the supplementary materials. Furthermore, we would like to highlight that the complete code, encompassing all processing steps, is publicly accessible via the link provided in the manuscript to ensure optimal transparency.

  1. Detailed Results Analysis: Conducting a more thorough analysis of the specific differences in performance metrics can clarify which models excel under what conditions.

  1. Additional Evaluation Metrics: Including other evaluation metrics (e.g., F1 score, recall) can provide a more comprehensive assessment of the models' performance.

We are pleased to provide the additional metrics for model performance. These can now be found in the supplementary materials available in Table S7.

Finally, we would like to highlight that MDPI Style correction service was now used to enhance the final quality in terms of language of the manuscript.

Thank you once more for your insightful remarks, which enrich the ongoing discourse and progress in our field.

Reviewer 2 Report

Comments and Suggestions for Authors

It is my pleasure to review the manuscript. The authors reported on “Enhancing Preoperative Outcome Prediction: A Comparative Retrospective 

Case-Control Study on Machine Learning versus the ISEG Risk Model for 

Predicting 90-Day Mortality in Oncologic Esophagectomy ”.  I think this is a novel article, but there are some corrections that need to be made.

Major comment

  1. A breakdown of the tumor stages listed in Table S1 should be included in Table 1.

  1. The type of complication resulting in death should also be stated.

  1. The extent of surgical dissection, radiotherapy dose should be mentioned if it is known, and if not, it should be mentioned in the limitation.

Author Response

Dear Sir or Madame,

We would like to express our utmost appreciation for your valuable comments and wish to sincerely respond to your remarks accordingly.

  1. “A breakdown of the tumor stages listed in Table S1 should be included in Table 1.”

We appreciate this valuable recommendation, which we have delightly implemented into the manuscript. Hence, the tumor stage has been moved from the supplementary table 1 to the main table of the main body.

  1. The type of complication resulting in death should also be stated.

We fully agree that highlighting the complications related to 90-day mortality is of crucial importance. Consequently, we have detailed the leading types of complications (sepsis, hemorrhagic shock, and myocardial infarction) for both centers within the corresponding paragraphs of the results section..

  1. The extent of surgical dissection, radiotherapy dose should be mentioned if it is known, and if not, it should be mentioned in the limitation.

We deeply appreciate this insightful remark. Recognizing the potential impact of neoadjuvant treatments, we have included the total number of chemotherapy cycles in our analysis. However, based on your recommendation, we have noted in the limitations section that potential reductions in individual chemotherapy cycles and variations in radiation doses were not considered. This omission is primarily due to the sparse documentation of such details, particularly for cases prior to 2015, resulting from changes in the hospital information system.

Finally, we would like to highlight that MDPI Style correction service was now used to enhance the final quality of the manuscript.

Thank you once again for your insightful comments, which contribute to the ongoing dialogue and advancement in our field.

Round 2

Reviewer 2 Report

Comments and Suggestions for Authors

The presented manuscript is revised adequately.

I appreciate the authors for their contributions.